# Functional Role of Extracellular Vesicles in Skeletal Muscle Physiology and Sarcopenia: The Importance of Physical Exercise and Nutrition

**DOI:** 10.3390/nu16183097

**Published:** 2024-09-13

**Authors:** Mauro Lombardo, Gilda Aiello, Deborah Fratantonio, Sercan Karav, Sara Baldelli

**Affiliations:** 1Department for the Promotion of Human Science and Quality of Life, San Raffaele Open University, Via di Val Cannuta, 247, 00166 Rome, Italy; mauro.lombardo@uniroma5.it (M.L.); gilda.aiello@uniroma5.it (G.A.); 2Department of Medicine and Surgery, LUM University, S.S. 100 Km 18, 70100 Casamassima, Italy; fratantonio@lum.it; 3Department of Molecular Biology and Genetics, Çanakkale Onsekiz Mart University, Canakkale 17000, Türkiye; sercankarav@comu.edu.tr; 4IRCCS San Raffaele Roma, 00166 Rome, Italy

**Keywords:** skeletal muscle, sarcopenia, myogenesis, extracellular vesicles, physical exercise, nutrition

## Abstract

Background/Objectives: Extracellular vesicles (EVs) play a key role in intercellular communication by transferring miRNAs and other macromolecules between cells. Understanding how diet and exercise modulate the release and content of skeletal muscle (SM)-derived EVs could lead to novel therapeutic strategies to prevent age-related muscle decline and other chronic diseases, such as sarcopenia. This review aims to provide an overview of the role of EVs in muscle function and to explore how nutritional and physical interventions can optimise their release and function. Methods: A literature review of studies examining the impact of exercise and nutritional interventions on MS-derived EVs was conducted. Major scientific databases, including PubMed, Scopus and Web of Science, were searched using keywords such as ‘extracellular vesicles’, ‘muscle’, ‘exercise’, ‘nutrition’ and ‘sarcopenia’. The selected studies included randomised controlled trials (RCTs), clinical trials and cohort studies. Data from these studies were synthesised to identify key findings related to the release of EVs, their composition and their potential role as therapeutic targets. Results: Dietary patterns, specific foods and supplements were found to significantly modulate EV release and composition, affecting muscle health and metabolism. Exercise-induced changes in EV content were observed after both acute and chronic interventions, with a marked impact on miRNAs and proteins related to muscle growth and inflammation. Nutritional interventions, such as the Mediterranean diet and omega-3 fatty acids, have also shown the ability to alter EV profiles, suggesting their potential to improve cardiovascular health and reduce inflammation. Conclusions: EVs are emerging as critical mediators of the beneficial effects of diet and exercise on muscle health. Both exercise and nutritional interventions can modulate the release and content of MS-derived EVs, offering promising avenues for the development of novel therapeutic strategies targeting sarcopenia and other muscle diseases. Future research should focus on large-scale RCT studies with standardised methodologies to better understand the role of EVs as biomarkers and therapeutic targets.

## 1. Introduction

All cell types are able to secrete extracellular vesicles (EVs) that can be isolated from different biological fluids using various methods [1]. To date, they are considered fundamental transporters in proximal and distal intercellular communication. In fact, their greatest capacity is to transport different biological loads that influence physiological and pathological processes. In this regard, many studies have been conducted on the relationship between EVs and chronic diseases such as cardiovascular diseases, diabetes, and cancers. EVs are gaining significant attention in recent research across various diseases due to their crucial roles in disease mechanisms and potential therapeutic applications. In cardiovascular research, EVs are implicated in disease progression and therapy. EVs from endothelial cells can transport pro-inflammatory and pro-thrombotic factors, exacerbating atherosclerosis and contributing to plaque instability [2]. Platelet and macrophage-derived EVs play roles in thrombosis and myocardial infarction by promoting clot formation and inflammation [3,4]. EVs from cardiomyocytes are being explored as non-invasive biomarkers for heart failure, offering insights into cardiac damage and disease severity [5]. Their therapeutic potential includes delivering protective factors to damaged tissues or modulating immune responses.

In cancer research, EVs are critical for tumor biology and metastasis. They can facilitate cancer spread by carrying enzymes that degrade extracellular matrix components and help tumors evade immune surveillance by transporting immunosuppressive molecules [6]. EVs are also emerging as valuable cancer biomarkers, reflecting the molecular characteristics of tumors [7]. Advances in EVs analysis technologies have improved the identification of cancer-specific markers and their use in targeted drug delivery or as therapeutic agents [8].

In diabetes research, EVs are recognized for their roles in disease mechanisms and potential as diagnostic tools. In type 1 diabetes, EVs from beta cells and immune cells may contribute to autoimmune destruction and inflammation [9]. In type 2 diabetes, EVs from adipocytes and muscle cells influence insulin resistance and systemic inflammation [10]. EVs are emerging as biomarkers for monitoring disease progression and therapeutic responses, and research is exploring their use in delivering therapeutic agents or modulating immune and metabolic responses. Overall, EVs are proving to be valuable in understanding and managing cardiovascular diseases, cancer, and diabetes, with ongoing research aiming to translate these findings into clinical applications.

EVs derived from immune cells, such as macrophages and dendritic cells, can carry pro-inflammatory cytokines like TNF-α and IL-6, which promote inflammation by signaling to other cells [11]. They also influence the migration and activation of immune cells, further amplifying the inflammatory response. On the other hand, EVs can also transport anti-inflammatory molecules like transforming growth factor-β (TGF-β), which help resolve inflammation and promote tissue repair. Therefore, EVs can act as either pro-inflammatory or anti-inflammatory agents, depending on their molecular content and the surrounding environment [12].

In addition to their role in inflammation, EVs are closely linked to oxidative stress. EVs can transport oxidative stress markers, including oxidized lipids and proteins, between cells, thereby spreading oxidative stress. EVs derived from cells under oxidative conditions often contain high levels of reactive oxygen species (ROS), which can induce oxidative damage in recipient cells [13]. However, EVs can also carry antioxidant enzymes like superoxide dismutase (SOD), which help mitigate oxidative stress, highlighting their dual role in both exacerbating and alleviating oxidative stress [14]. For these reasons, EVs are now used in multiple types of anti-inflammatory therapy, in vaccination, and in drug administration [15]. Alongside these advances, the ability of skeletal muscle (SM) to secrete myokines (defined as “muscle” cytokines and peptides that mediate communication between SM and other organs) has been highlighted, modifying the activity of other organs such as bone tissue, adipose tissue, and the pancreas [16,17,18]. Thus, the SM is also seen as a secretory organ capable of mediating autocrine, paracrine, and endocrine effects. In addition to secreting myokines, the SM also appears to carry out another type of communication: it is able to produce EVs that represent new paracrine and endocrine signals [19].

It has recently been demonstrated that physical exercise (PE) induces a greater and rapid production of SM-EVs in the extracellular environment [20]. These EVs would have the fundamental role of inter-tissue communication during PE by releasing different types of myokines into circulation [21]. Together with myokines, PE causes the release of RNA, miRNA, and mtDNA into the bloodstream, which presumably could represent the mediators of the beneficial health effects induced by PE. From this information, we can deduce the importance that EVs currently have in the modulation of metabolism, cellular differentiation, and regeneration at the SM level [22,23,24]. Moreover, in recent years, interest has emerged in plant-derived extracellular vesicles EVs (PDEVs), which, containing molecules of antioxidant origin, are considered therapeutic agents against many pathologies, including muscular ones [25]. Nutrition plays a crucial role in this context, as dietary interventions can influence the production, composition, and function of both endogenous and plant-derived EVs. Specific nutrients and bioactive compounds, such as polyphenols, fatty acids, and vitamins have been shown to modulate the release of EVs, influencing their load and biological activity. This interaction between diet, EVs and muscle health opens new perspectives for nutritional strategies to improve muscle function and overall metabolic health.

The aim of this review is therefore to provide current knowledge on the role of EVs/PDEVs and their contents at the level of SM in both physiological and pathological conditions. Furthermore, we will explore the potential implications of nutritional interventions on the regulation and function of these vesicles, highlighting the interaction between diet, exercise, and cellular communication.

## 2. Skeletal Muscle as an Endocrine Tissue

SM accounts for approximately 40 percent of the total body mass and is one of the largest organs in the body in terms of weight. In addition to its main function of locomotion, SM serves as a protein store, regulates glucose metabolism, and plays a key role in thermoregulation [26]. The SM is highly adaptive and responds to external stimuli such as PE, inactivity, inflammation, and nutrition, leading to changes in the size and structure of muscle fibers. These adaptations are mediated by the release of signaling molecules known as myokines, which include cytokines and hexokines. Importantly, many of these signaling molecules are transported through EVs, allowing them to exert their effects on peripheral organs (e.g., pancreas, adipose tissue, bone) or on the muscle itself through autocrine, paracrine, and endocrine pathways [27]. In recipient tissues, EVs-transported myokines can regulate physiological and pathological processes, such as metabolic capacity, hormone secretion, and even cognitive function. For instance, IL-6, a known myokine, can be encapsulated in EVs and has been shown to influence glucose uptake and fatty acid oxidation through the activation of AMPK [28]. The identification of myocinoma has provided important insights into how SM communicates with other organs, particularly through EV-mediated pathways. Understanding these interactions is critical to understanding the broader role of EVs in intertissue communication and metabolic regulation. In addition to physical exercise, specific dietary components such as omega-3 fatty acids and polyphenols have been shown to influence the release and composition of muscle-derived EVs, thereby modulating their endocrine functions. For instance, polyphenols can enhance the anti-inflammatory properties of EVs, contributing to improved metabolic responses and muscle health [1,29].

SM functions as an endocrine organ by producing and releasing EVs that carry various molecules, including microRNAs (miRNAs), tRNAs, mRNAs, proteins, and lipids, to recipient cells. These EVs facilitate intercellular communication in both physiological processes like myogenesis and pathological conditions such as atrophy and sarcopenia. Muscle cells secrete exosomes, a type of EVs, which influence muscle metabolism and disease. Research has identified miRNAs within these exosomes, particularly myomiRNAs like miRNA-1, miRNA-133a, miRNA-133b, and miRNA-206, which play key roles in muscle function. Studies have shown that treatment with guanosine-5-triphosphate increases the production of myomiRNAs, which are then released into circulation. These circulating and muscle-derived miRNAs act as autocrine and paracrine factors, influencing muscle growth and function. Research by Sork and colleagues demonstrated that miRNAs within myoblast-derived EVs regulate key pathways related to muscle physiology, mass, immunity, neuromuscular junctions, and calcium signaling. Importantly, the packaging of miRNAs into muscle EVs is selective, with some miRNAs present in myoblasts but not in myotubes. In addition to miRNAs, muscle EVs also contain tRNAs, whose functions are still unknown, as well as mRNAs encoding various receptors, ion channels, and transcription factors [30].

Lipids are a significant component of EVs in SM. Molecules like sphingomyelin, hexosylceramide, phosphatidylinositol, and free cholesterol found in muscle EVs could serve as biomarkers for muscle diseases. Recent studies highlight bioactive lipids such as ceramide (Cer) and diacylglyceride (DG) in muscle EVs, which might influence cell communication. For instance, muscle under hypertrophic stimulation releases EVs that increase lipolysis in adipose tissue. Muscle EVs also contain fatty acids like palmitic acid, which are used in plasma membrane construction [31,32]. These EVs can impact lipid metabolism in fat tissue by transporting enzymes linked to lipid processing, potentially altering lipid metabolism without directly changing gene expression.

Although proteins are present in low quantities within muscle EVs, those that are identified play key roles in metabolism and cell differentiation. Specifically, muscle EVs contain proteins involved in protein synthesis, folding, trafficking, RNA modifications, translation, and amino acid metabolism [33]. They also include proteins linked to gluconeogenesis, pyruvate metabolism, and carboxylic acid biosynthesis, highlighting the importance of muscle EVs in regulating muscle metabolism [34]. Advances in mass spectrometry have allowed for detailed analysis of these proteins, revealing a high content of post-translationally modified membrane proteins, including phosphorylated tetraspanins and glycosylated ligand-receptor pairs.

## 3. Biogenesis, Isolation, and Characterization of EVs

EVs originate from the plasma membrane or endosome and play a crucial role in intercellular communication by carrying functional molecules like nucleic acids, proteins, and lipids. EVs are classified into three main types based on their biogenesis: exosomes, microvesicles, and apoptotic bodies (Figure 1). These EVs can be isolated through various methods, such as precipitation, filtration, ultracentrifugation, density gradients, or a combination of these techniques, to ensure their purity and accurate classification. Significant advancements have been made in isolating EVs from SM tissue through modern techniques. While differential ultracentrifugation is still widely used, combining it with density gradient separation or size exclusion chromatography (SEC) enhances yield and purity. These methods, which include fluorescence/confocal microscopy for non-destructive analysis and real-time monitoring of muscle-specific miRNAs, transmission electron microscopy (TEM) for high-resolution imaging, atomic force microscopy (AFM) for detailed surface morphology, and several others like nanoparticle tracking analysis (NTA) and mass spectrometry (MS), allow for a detailed study of EVs, including their proteomic and lipidomic profiles [35]. It is also important to consider how dietary interventions can impact the biogenesis and release of EVs. Nutritional elements such as vitamins and polyunsaturated fats can modulate the lipid composition of EV membranes, influencing their stability and signaling properties. This interaction between diet and EV composition may be critical in understanding how dietary strategies can enhance or hinder the therapeutic use of EVs [21,36]. Additionally, specific protein markers, such as CD63, CD81, and TSG101, help classify EVs according to guidelines set by the International Society of Extracellular Vesicles (ISEV) [37].

## 4. EVs in the Myogenesis Process

EVs play a significant role in modulating SM homeostasis and differentiation. Studies have shown that the miRNA content within muscle EVs changes notably during myogenic differentiation. These miRNAs regulate various processes such as contractility, reactivity, proliferation, and differentiation of muscle cells. Specifically, miRNA-133a, -206, and -1 influence gene expression and impact the development and function of myoblasts and myotubes [38]. Additionally, factors like MADS, SRF, and MEF2 are crucial for muscle cell division, differentiation, and overall muscle development [39].

Several molecules, particularly myomiRNAs, play a crucial role in muscle regulation and differentiation. Key myomiRNAs include miRNA-1, miRNA-133a, miRNA-133b, miRNA-206, miRNA-208a, miRNA-208b, miRNA-486, and miRNA-499 [40]. Their expression is controlled by muscle-specific transcription factors like MRFs, MYOD1, MYF5, MYF6, MYOG, MEF2, SRF, and MKL1. Other important miRNAs in myogenesis include miRNA-24, miRNA-29, miRNA-125, miRNA-181, miRNA-214, miRNA-221/222, miRNA-322/424, miRNA-503, and miRNA-675. Among them, miRNA-1, miRNA-133a/b, and miRNA-206 are the most studied. Specifically, miRNA-1, regulated by mTOR, MYOD, and YY1, can form regulatory loops and modulate HDAC4, a repressor of MEF2 [41]. miRNA-1 and miRNA-206 regulate PAX7, which in turn enhances the activity of ID2, a protein that promotes muscle cell differentiation and inhibits myoblast proliferation [42]. Research by Chen and colleagues found that miRNA-1 and miRNA-133 are transcribed from the same chromosomal region but function differently: miRNA-1 inhibits myoblast proliferation, while miRNA-133 promotes it by repressing SRF. Together, these miRNAs form a regulatory loop controlling muscle cell proliferation and differentiation [41]. Additionally, miRNA-206 supports smooth muscle differentiation by negatively regulating DNA polymerase α, which inhibits DNA synthesis. Disrupting the activity of miRNA-1 and miRNA-206 highlights their crucial role in proper muscle development [43,44].

EVs not only contain and release crucial miRNAs that regulate myogenesis but also secrete essential factors for muscle development, including insulin-like growth factor-1 (IGF-1), TGF-β3, vascular endothelial growth factor, and fibroblast growth factor 2 (FGF2) [45]. Proteomics and mass spectrometry studies have shown that the protein composition of myoblast-EVs varies with their differentiation state. Specifically, research by Forterre and collaborators revealed that myoblast-EVs have higher levels of myogenin compared to cyclin D, suggesting that EVs from myotubes can influence myoblast differentiation. Baci and colleagues confirmed this by demonstrating that EVs can shuttle molecules that affect muscle regeneration and disease. They developed a protocol using EVs to enhance the differentiation of pluripotent stem cells (iPSCs) into myotube-like cells, improving differentiation and fusion rates by up to 70%. Furthermore, injecting differentiated iPSCs into the tibialis muscle of α-sarcoglycan knockout immunodeficient mice showed that these cells can integrate into host myofibers, highlighting the potential of this approach for regenerative medicine. However, the use of muscle EVs as biomarkers for physiological, pathological, or aging conditions remains unclear, necessitating further research to explore their properties and potential as therapeutic targets in conditions like sarcopenia.

## 5. Sarcopenia and Muscle EVs

A critical global public health issue is the gradual decline in muscle strength and mass that happens with aging, known as sarcopenia. Since this condition often overlaps with other age-related pathologies, it is not always easy to study it as a unique phenomenon. Nutritional interventions, particularly PE have proven to be important strategies for managing sarcopenia. To date, there are still no drugs that can prevent, delay, or treat sarcopenia, not to mention biomarkers that could be used in both clinical and research settings [46].

It is known that sarcopenia is characterized by significant mitochondrial inflammation and dysfunction. High levels of the proinflammatory cytokine IL-6 have been found in subjects suffering from sarcopenia [47]. The implementation of the inflammatory process causes the release of damage-related molecular pattern molecules (DAMPs), such as nuclear and mitochondrial DNA (mtDNA), some proteins, reactive oxygen species (ROS), and other molecules [48]. Next to this, the mitochondrial quality control (MQC) machinery seems to be totally altered in patients with sarcopenia. These alterations cause poor regenerative capacity of the SM. In fact, in physiological conditions, the muscle, through the satellite cells, can regenerate itself after an injury or a disease. However, in patients with sarcopenia the number and function of satellite cells decrease drastically, thus not guaranteeing the regenerative process. Being able to modulate the activity of these cells is currently considered one of the possible treatments for sarcopenia. The molecular mechanism underlying the inability of satellite cells to regenerate SM was demonstrated by Shao and colleagues [49]. The authors’ results show how the release of miRNA-690 from muscle EVs in atrophic muscles inhibits the myogenic activity of satellite cells by targeting myocyte growth factor (MEF2). The development, differentiation, and fusion of myoblasts in the SM is dependent on MEF2 [50]. It therefore appears that EVs during sarcopenia transfer specific miRNAs that block/delay myogenesis and muscle cell regeneration. Therefore, in order to increase mass and muscle strength and counteract sarcopenia, it would be desirable to consider miRNA-690 as a therapeutic target. Recent studies suggest that nutritional interventions, particularly those rich in antioxidants and anti-inflammatory nutrients, can modulate the miRNA content within muscle EVs in patients with sarcopenia. For example, diets high in omega-3 fatty acids or polyphenol-rich foods have been associated with a reduction in pro-inflammatory miRNAs, potentially slowing the progression of sarcopenia [46].

Another group of researchers showed how miRNA-1 and -206 could be used against muscle damage observed during sarcopenia as well as in SM regeneration. As previously mentioned, miRNA-1 and -206 are inversely correlated to the expression of PAX7 (which maintains satellite cells in a quiescent state) [51]. Their levels have been observed to be strongly downregulated in damaged/sarcopenic muscles. Today it is therefore hypothesized that these miRNAs could be infused directly into the skeletal muscle as a “reparative treatment” against muscle damage induced by sarcopenia. These data were also confirmed by other studies showing how direct infusion of miRNA-1, miRNA-133, and miRNA-206 stimulates MRFs expression and consequently recovery in rats with muscle damage [52]. In a very interesting study, the authors show high levels of miRNA-Let-7 in the muscles of elderly subjects. This increased expression correlates with a reduction in muscle self-renewal. In fact, some studies associate the expression of miRNA-Let-7 with the regulation of the cell cycle. In particular, miRNA-Let-7 causes a decrease in CDK6, CDC25A, and CDC34, which are considered positive regulators of muscle cell proliferation. These data suggest that in sarcopenic muscle the regulation of the cell cycle and the quantity of satellite cells is lower than in healthy muscle, but this can be modulated through a miRNA-Let-7-dependent mechanism [53] (Figure 2).

Another aspect that plays a fundamental role in sarcopenia is mitochondrial dysfunction. Fusion, fission, and MQC processes are altered, along with the accumulation of oxidative stress on proteins, lipids, and mtDNA, contributing to mitochondrial dysfunction. It has been demonstrated that some miRNAs, such as miR-1, miR-133, and miR-206, when infused directly into damaged and sarcopenic muscles of rats are able to accelerate the recovery of function; this suggests a role for miRNAs in the reparative procedure involving the damage to the muscle. Mitochondrial components (e.g., ATP5A (complex V), NDUFS3 (complex I), and SDHB (complex II) and mtDNA) have been found in muscle EVs of sarcopenic subjects, which can activate the inflammatory pathway by interacting with several receptors/systems, including TLRs, family pyrin domain-containing 3 (NLRP3) inflammasome, and cGAS-STING DNA sensing system [54]. Other studies demonstrate the presence of pro-inflammatory proteins in the EVs of sarcopenic subjects, such as FGF-21, HGF, IL-12B, PD-L1, PRDX3, and STAMBP. Furthermore, high levels of mtDNA were found in the EVs of elderly subjects with two particularly expressed genes NADH dehydrogenase 2 (MT-ND2) gene region (Mito_4625) and Cytochrome c oxidase subunit 2 (COX2) gene region (Mito_7878) [55], identifying the mtDNA as one DAMPs. These studies show that EVs can transport DAMPs such as mtDNA as well as inflammatory proteins in patients with sarcopenia and present a way to guide the development of biomarkers and interventions for the treatment of sarcopenia, thus identifying the molecules underlying sarcopenia. Moreover, it has been demonstrated that some miRNAs can regulate mitochondrial homeostasis, substantially contributing to the dysfunction of this organelle and consequently to the development of sarcopenia. In particular, miRNA-696 negatively regulates fatty acid oxidation and mitochondrial function by acting directly on PGC-1α transcription [56]. Another miRNA that appears to have a fundamental role in mitochondrial and SM dynamics is miRNA-133/a. Its deficiency is associated with a decrease in PGC-1α and NRF1 levels. This causes dysfunction and a decrease in mitochondrial mass, an increase in ROS, and inflammation [57,58].

Muscle atrophy that occurs during sarcopenia is characterized by loss of MQC homeostasis. When MQC acts correctly, it guarantees the correct homeostasis and functionality of the mitochondrion. When working at full capacity, this process maintains mitochondrial quality in the cell. However, a decrease in MQC capacity occurs during aging and degenerative diseases. In recent years, the relationship between EVs in the MQC, lysosomal, and mitochondrial compartments for sarcopenia has been highlighted [59]. Indeed, the presence of mitochondrial constituents in EVs serves as indirect evidence of a crosstalk between mitochondria and the endolysosomal system. Such communication can occur even without depolarization of the mitochondrion or in the absence of genes related to autophagy, indicating that mitochondrial-EVs delivery to lysosomes for degradation complements mitophagy for MQC. During sarcopenia, the link between mitochondria, lysosomes, and EVs secretion is strongly altered and could be the cause of the loss of MQC homeostasis in SM, causing inflammation and progressive loss of strength and muscle mass [60]. Understanding the molecular mechanisms of MQC failure during sarcopenia would be very useful in identifying novel biological targets for anti-aging interventions. In the muscles of patients with sarcopenia, the content and type of miRNAs significantly change. In the muscle and blood of sarcopenic subjects, 13 miRNAs have been identified that can potentially be modulated miRNA-10a-3p, -19a, -21, 34a, -92a-3p, 185-3p, 194-3p, -203a-3p, -326, -424-5p, -532-5p, -576-5p and -760. These miRNAs are characteristic of some particular traits of sarcopenic subjects, such as weakness, poor resistance and energy, slowness, and low levels of PE. Modulation of the content of these miRNAs affects sarcopenia. miRNA-181a binds to the 3′ untranslated region of Sirt1 and influences aging, apoptosis, and inflammation. Its overexpression causes a decrease in myotube diameter, while its downregulation causes an increase [61]. Overexpression of miRNA-455-3p significantly increased the diameter of cultured myotubes by suppressing the expression levels of PITX1 and RXRB, which are involved in muscular dystrophy and aging [62]. Furthermore, the differential expression of miRNAs can modulate the MQC and autophagy/lysosome systems, causing substantial changes in protein/organelle degradation. In particular, MuRF1 and MAFbx, two E3 ubiquitin ligases important for protein degradation during muscle atrophy were found to be regulated by miRNA-23a, miRNA-27a, and miRNA-351. Overexpression of miRNA-29c can increase muscle mass by inhibiting MuRF1 [63]. miRNA-486, which targets FoxO1 and phosphatase tensin homolog (PTEN), suppresses signaling that leads to protein degradation and subsequently muscle atrophy [64]. It has been reported that some miRNAs have important functions also at the level of the regulation of proteins implicated in the UPS and autophagy/lysosomes system (PI3K/AKT/mTOR and TGF-β/myostatin/BMP pathways) indispensable in myogenic differentiation. An example is that of miR-125b, miR-223, and miR-199-3p, they modulate the PI3K/AKT/mTOR signaling pathway and the rhythm of protein synthesis by operating through insulin-like factor (IGF)-2 and IGF-1, respectively [65,66]. miRNA-21 also regulates SM development by targeting transforming growth factor-beta (TGFβI) and inhibiting PI3K/AKT signaling [67]. Although the molecular targets of these miRNAs have been identified, much remains to be understood to regulate protein synthesis and mitochondrial homeostasis during sarcopenia. These studies show that a decrease in muscle quantity and function associated with sarcopenia also causes a decrease in the quality of life. To date, many studies are still underway to better understand this phenomenon and to identify one or more therapeutic interventions to improve the lives of individuals with sarcopenia (nutrition and PE). Recent data underline the importance and potential of EVs secreted by myotubes: by transporting miRNAs and functional molecules, they could not only regulate the functionality of the SM but also contribute to research strategies on sarcopenia.

## 6. The Impact of Diet, Food, and Exercise on EVs

Recent studies have shown how diet, specific foods, and exercise can influence the release, composition, and function of EVs, thus affecting overall health.

### 6.1. Methodology

This study reviewed the existing literature on the role of EVs in skeletal muscle function, focusing on the effects of exercise and nutrition. Studies published in English that reported data on the impact of EVs on SM physiology and examined the effects of exercise and nutritional interventions on EVs release and function were included in the review.

Major scientific databases, including *PubMed*, *Scopus*, and *Web of Science* were consulted to identify relevant articles published up to July 2024. The search terms used included combinations of keywords such as ‘extracellular vesicles’ AND ‘muscle’ OR ‘exercise’ OR ‘nutrition’ OR ‘sarcopenia’. The data extracted from the selected studies included this study design, study population, duration of the intervention, outcome measures related to EVs, and primary outcomes. The collected data were summarized in a summary table and analyzed qualitatively to identify common trends and gaps in the existing literature. The effects of EVs on various aspects of SM function and their therapeutic potential were discussed in light of the available evidence. This study selection process is illustrated in Figure 3, which outlines the stages from initial identification to final inclusion, including the reasons for study exclusion at each stage.

### 6.2. Impact of Diet on EVs

The effects of dietary patterns, individual foods, and dietary supplements on EVs are summarized in Table 1.

The Mediterranean diet (MD) has been shown to significantly influence EVs. Chiva-Blanch et al. found that CD142+/CD61+, CD146+, and CD45+ microparticles can predict cardiovascular events in high-risk patients on a nut-enriched MD. This suggests that these microparticles could serve as predictive biomarkers for cardiovascular risk [68]. In another study, the same research group found that the MD reduces the release of pro-thrombotic microvesicles in asymptomatic but high cardiovascular-risk individuals, indicating a potential protective effect through modulation of EV [69]. Kwon et al. explored the influence of the MD on miRNA expression in EVs of breast cancer survivors. The results showed significant changes in miRNA expression, suggesting that the MD could influence cellular communication and biological processes [70].

Two other papers evaluated the impact of diet on EVs. Eitan et al. examined the effect of protein restriction in the diet on leptin and insulin signaling markers in plasma EVs. They found that protein restriction alters these markers, indicating that diet composition may modulate important metabolic pathways through EVs [71]. Weech et al. found that substituting dietary saturated fat with unsaturated fats leads to an increase in circulating endothelial progenitor cells and a reduction in microparticles [72].

### 6.3. Influence of Specific Foods on EVs

Several studies have highlighted the impact of specific foods on EVs and their effects on various aspects of health. Zhang et al. found that an oat-enriched diet reduced the inflammatory state as assessed by circulating microparticles in patients with type 2 diabetes. This suggests that oats may have anti-inflammatory properties that are reflected in EVs levels [73]. Gröne et al. demonstrated that cocoa flavanols improve the functional integrity of the endothelium in both the young and the elderly. This observation suggests that cocoa consumption may contribute to maintaining vascular health through the modulation of EVs [74]. Ammollo et al. found that grape consumption reduces thrombin generation and improves plasma fibrinolysis by reducing procoagulant microparticles, indicating a potential antithrombotic benefit [75]. López de Las Hazas et al. observed that dietary supplementation with walnuts modifies exosomal miRNAs in elderly subjects. This study suggests that walnuts may influence gene regulation through EVs, potentially improving metabolic health [76]. Horn et al. (2014) found that dietary flavanol intervention lowers the levels of endothelial microparticles in patients with coronary artery disease [77]. Finally, Bryl-Górecka et al. observed that blueberry supplementation reduces platelet and endothelial microparticles in patients with myocardial infarction, suggesting a protective effect of blueberries on cardiovascular health through modulation of EVs [78]. Yang et al. demonstrated that the consumption of a Leuconostoc holzapfelii-enriched synbiotic beverage significantly alters the composition of the microbiota and microbial EVs [79].

### 6.4. Effects of Nutritional Supplements on EVs

Several studies have examined how nutritional supplementation can influence EVs and general health. These studies vary in the methods of intervention and the results observed, but can be grouped into three main categories: vascular health, inflammatory modulation, and general well-being.

With regard to vascular health, several studies have examined the effect of specific supplements on the profile of microparticles and endothelial cells. For instance, Wu et al. investigated the effects of fish oil supplementation on endothelial progenitor cells and microparticles in individuals at moderate cardiovascular risk. Their findings revealed an increase in endothelial progenitor cells and a decrease in microparticles, suggesting that fish oil may improve vascular health by modulating EVs [80]. Similarly, Burnley-Hall et al. studied the effect of nitrate supplementation on the reduction of platelet-derived microparticles in patients with coronary artery disease receiving clopidogrel therapy. They observed a significant reduction in platelet-derived microparticles, indicating that nitrate may enhance the efficacy of clopidogrel in preventing thrombotic complications [81]. Further supporting the relevance of fatty acids, Bozbas et al. observed that n-3 polyunsaturated fatty acids alter the number, fatty acid profile, and clotting activity of circulating and platelet-derived EVs [29]. However, Phang et al. found that circulating CD36+ microparticles are not altered by docosahexaenoic or eicosapentaenoic acid supplementation [82], though they later demonstrated that acute supplementation with eicosapentaenoic acid reduces platelet microparticle activity in healthy subjects [36]. In contrast, Chiva-Blanch et al. noted that one year of ω-3 polyunsaturated fatty acid supplementation does not reduce circulating prothrombotic microvesicles in elderly subjects following a myocardial infarction [83]. Moreover, Endothelial microparticles (EMP) are small fragments originating from the endothelial cell membrane. They are released into the circulation in response to endothelial activation, injury, proliferation, or apoptosis. A combination of oral nutraceuticals significantly improved the cholesterol profile by reducing EMP by 16% and hsCRP by 41% [84]. Regarding the modulation of inflammation, several studies have explored the impact of specific supplements, particularly focusing on the role of vitamin C and omega-3 fatty acids in influencing microparticle levels under different inflammatory conditions. Weisshaar et al. studied the effect of high-dose vitamin C on LPS-induced microparticle increase, reporting that vitamin C does not mitigate microparticle formation during acute inflammation [85]. In contrast, Yang et al. demonstrated that ascorbic acid supplementation decreases microparticle elevation and neutrophil activation after scuba diving [86]. In terms of overall health, including skin, gut, and metabolic balance, several studies have focused on the effects of specific supplements on EVs. Kim et al. found that Lactobacillus plantarum CJLP55 improves clinical conditions, skin conditions, and urinary bacterial EVs in patients with acne vulgaris. This suggests that the probiotic may modulate EVs to improve skin health and microbial balance [87]. Similarly, Shin et al. evaluated the safety and efficacy of ID-JPL934 in improving lower gastrointestinal symptoms and found significant benefits, indicating that this supplement may positively influence EVs and gut health [88]. Furthermore, Nederveen et al. found that a multi-ingredient supplement aids in weight loss and enhances body composition in overweight and obese individuals [89].

**Table 1 nutrients-16-03097-t001:** Summary of Human Studies on the Impact of Diet and Dietary Supplements on Extracellular Vesicles (EVs) and Related Outcomes.

First Author	Year	Sample Size (n)	Study Design	Population	Duration	Intervention Type	Effects on EV	Other Outcomes	Ref.
Phang, M.	2012	30	Comparative Study	Healthy subjects	24 h	EPA	↓ platelet microparticle activity, DHA did not	Both EPA and DHA ↓ platelet aggregation, gender-dependent effects observed	[36]
Weisshaar, S.	2013	14	RCT	Healthy male subjects	6 h	Vitamin C	=MP formation post-LPS exposure	↑ MP levels during acute systemic inflammation	[85]
Horn, P.	2014	16	Clinical Trial	Patients with coronary artery disease	1 month	Flavanol	↓ levels of CD31+/41− and CD144+ endothelial EVs	Improved endothelial function	[77]
Wu, S-Y.	2014	84	RCT	Subjects at moderate risk of cardiovascular disease	8 weeks	Fish-oil	↑ numbers of EPCs, ↓ numbers of EMPs	No significant effects on blood pressure, plasma lipids, or plasma glucose	[80]
Zhang, X.	2014	22	RCT	Patients with type 2 diabetes	8 weeks	Oat-enriched diet	↓ concentrations and proportions of fibrinogen- and tissue factor-related platelet and monocyte microparticles	Improved inflammatory status assessed by microparticle concentrations	[73]
Yang, M.	2015	14	Controlled Clinical Trial	Healthy male divers	6 days	Ascorbic acid	diminished microparticle elevations post-SCUBA diving	↓ neutrophil activation, no effect on intravascular bubble production	[86]
Chiva-Blanch, G.	2016	50	RCT	High cardiovascular-risk individuals	1 year	MD + nuts	↓ levels of CD142+/CD61+/AV+, CD146+/AV+, and CD45+/AV+ microparticles in patients with no-CVE compared to CVE	Predictive model for future cardiovascular events with high accuracy	[68]
Phang, M.	2016	94	RCT	Healthy men and women	4 weeks	DHA, EPA	=CD36+ MPs	Cardioprotective effects of DHA and EPA do not act through CD36+ MP mechanism	[82]
Pirro, M.	2016	100	RCT	Patients with subclinical inflammation	3 months	Nutraceutical combination	↓ endothelial microparticles (−16%) and hsCRP (−41%)	Improved cholesterol profile (total and LDL cholesterol)	[84]
Ammollo, C. T.	2017	30	Comparative Study	Healthy volunteers	3 weeks	Grapes	↓ procoagulant microparticles	↓ thrombin generation and enhanced plasma fibrinolysis, sustained anticoagulant, and profibrinolytic effects	[75]
Eitan, E.	2017	38	RCT	Patients with prostate cancer awaiting prostatectomy	1 month	Dietary protein restriction	↑ levels of leptin receptor in total plasma EVs and L1CAM+ EVs, altered phosphorylation status of IRS1 in L1CAM+ EVs	Improved insulin and leptin sensitivity	[71]
Burnley-Hall, N.	2018	20	RCT	Patients with coronary artery disease on clopidogrel therapy	16 days	Nitrate supplementation	↓ circulating platelet-derived extracellular vesicles (CD41+ EVs)	↑ plasma RSNO levels, ↓ thrombin-receptor mediated platelet aggregation	[81]
Weech, M.	2018	190	RCT	Adults with moderate CVD risk	16 weeks	Replacement of dietary saturated fat with unsaturated fats	↓ endothelial microparticles (−47.3% for MUFA, −44.9% for n-6 PUFA) and platelet microparticles (−36.8% for MUFA, −39.1% for n-6 PUFA)	↑ endothelial progenitor cell numbers (+28.4% for MUFA)	[72]
Yang, J.	2019	21	Clinical Trial	Healthy Korean adults	4 weeks	*Leuconostoc holzapfelii*-enriched synbiotic beverage	Significant ↑ in species diversity of circulating urinary EVs	Lowered AST serum levels, particularly in subjects with starting levels > 40 UI/L	[79]
Bryl-Górecka, P.	2020	50	Open-Label Study	Patients with myocardial infarction	8 weeks	Bilberry	↓ platelet-derived microvesicles (PMVs) and endothelial-derived microvesicles (EMVs)	↓ endothelial EV release, Akt phosphorylation, and vesiculation-related gene transcription	[78]
Chiva-Blanch, G.	2020	155	RCT	High CVD individuals	1 year	MD	↓ prothrombotic microvesicle release compared to low-fat diet	Lower cell activation towards a pro-atherothrombotic phenotype, suggesting delayed CV complications	[69]
Gröne, M.	2020	39	Clinical Trial	Healthy young and elderly subjects	2 weeks	Cocoa flavanols	↓ concentrations of CD31+/41−, CD144+, and CD62e+ EMPs	Improved FMD and vascular function	[74]
Kwon, Y-J.	2020	16	Clinical	Breast cancer survivors	8 weeks	MD	42 EV miRNAs significantly differentially regulated (36 up-regulated, 6 down-regulated)	Improved BMI, waist circumference, fasting glucose, insulin, and HOMA-IR	[70]
Chiva-Blanch, G.	2021	156	RCT	Elderly subjects post-myocardial infarction	1 year	ω-3 PUFA	No significant modulation of prothrombotic microvesicle release from blood and vascular cells	↑ levels of various microvesicle subtypes in both ω 3 and placebo groups	[83]
Kim, M-J.	2021	28	RCT	Patients with acne vulgaris	12 weeks	*Lactobacillus plantarum* CJLP55	↓ prevalence of Proteobacteria and ↑ Firmicutes in urine bacterial EVs	Improved acne lesion count and grade, ↓ sebum triglycerides, ↑ skin hydration, and ceramide 2	[87]
López de Las Hazas, M.-C.	2021	211	RCT	Elderly subjects	1 year	Walnuts	Induced exosomal miRNAs (hsa-miR-32-5p and hsa-miR-29b-3p)	No major changes in exosomal lipids, nanoparticle concentration, or size	[76]
Shin, C. M.	2021	112	RCT	Patients with lower gastrointestinal symptoms	8 weeks	ID-JPL934	Significant ↑ in Lactobacillus johnsonii and Bifidobacterium lactis in feces post-treatment	Higher relief of overall gastrointestinal symptoms, ↓ abdominal pain, and bloating scores	[88]
Nederveen, J. P.	2023	55	RCT	Overweight and obese individuals	12 weeks	Multi-ingredient supplement	Significant ↓ in EVs-associated miRNA species miR-122 and miR-34a	Improved weight, fat mass, liver health, and metabolism	[89]
Bozbas, E.	2024	40	RCT	Individuals with moderate CVD risk	12 weeks	ω-3 PUFA	↓ numbers of circulating EVs, doubled *n*-3 PUFA content in EVs, ↓ EV capacity to support thrombin generation by >20%	=thrombus formation in ex vivo assay	[29]

This table summarizes studies on the effects of different interventions on extracellular vesicles (EVs) and related outcomes. ‘The arrows ‘↑’ and ‘↓’ indicate an increase or decrease in levels respectively, while the symbol ‘=’ indicates no significant change.’ Abbreviations used in the table: AST: aspartate aminotransferase; BMI: body mass index; CD: cluster of differentiation; CVD: cardiovascular disease; DHA: docosahexaenoic acid; EMP: endothelial microparticles; EPA: eicosapentaenoic acid; EV: extracellular vesicles; FMD: flow-mediated dilation; HOMA-IR: homeostasis model assessment for insulin resistance; hsCRP: high-sensitivity C-reactive protein; ID-JPL934: proprietary probiotic strain; LPS: lipopolysaccharide; MD: Mediterranean diet; miRNA: microRNA; MP: microparticles; MUFA: monounsaturated fatty acids; PUFA: polyunsaturated fatty acids; RCT: randomized controlled trial; RSNO: S-nitrosothiols; SCUBA: self-contained underwater breathing apparatus.

### 6.5. Exercise-Induced Changes in EVs

Exercise has been reported to have multisystem benefits in humans, including reducing cardiovascular risk, improving glycometabolic homeostasis, promoting weight loss, stimulating anabolic hormones, and preventing sarcopenia [90]. Moreover, it counteracts the muscular atrophy that accompanies aging by modulating myogenesis and protein metabolism [91]. In this type of muscle adaptation, which is necessary to maintain good health, EVs seem to play a fundamental role as they may mediate adaptive responses to exercise. Early studies demonstrate a change in the number and content of muscle EVs already after a single bout of exhaustive endurance exercise. Whitman and colleagues report a significant increase in different protein groups associated with circulating EVs after PE, which are then transferred to target cells, highlighting the biological changes that EVs can implement following PE [21]. A significant correlation was highlighted between aerobic capacity and plasma-isolated EVs (miRNA-133b and miRNA-181a-5p), which were found upregulated after acute PE [92]. miRNA-486-5p, miRNA-215-5p, and miRNA-941 levels were also found to be higher following regular PE. These miRNAs are connected to the IGF-1 molecular pathway involved in exercise-induced muscular and cardiac hypertrophy [93]. Exercise intensity also plays an important role in circulating EVs loading and release. This concept was highlighted by a study in which different exercise modalities differently modulated the load of miRNAs-126 and -133, two markers of muscle damage. 4 h of cycling at 70% anaerobic threshold stimulated plasma concentrations of miRNA-126 with no effect on miRNA-133.

On the contrary, at the end of a marathon, an increase of both miRNA-126 and -133 was observed. Differently, eccentric resistance training determines only the increase of miRNA-133 [94]. Moreover, Kargal et al., evaluated the considerable sex-specific effects of the training on EV miRNA profiles, with men showing larger variation than women in certain miRNAs associated with muscle growth and hypertrophy pathways. Both chronic and acute exercise influenced EVs characteristics, but the responses differed significantly between sexes. Men showed more pronounced changes in pathways related to muscle hypertrophy and growth after training, suggesting a gender difference in how training affects EVs signaling [95]. Therefore, the modality and intensity of PE are fundamental characteristics to take into consideration when studying the release of EVs from SM into the circulation. These data were also supported by other studies that show how some miRNAs in SM are modified after acute PE: Nielsen et al. show that 8 circulating miRNAs (miRNA-106a, miRNA-221, miRNA-30b, miRNA-151-5p, let-7i, miRNA-146, miRNA-652 and miRNA-151-3p) were downregulated after acute exercise. After 1–3 h of acute resistance exercise, miRNA-338-3p, miRNA-330-3p, miRNA-223, miRNA-139-5p, miRNA-143, and miRNA-1 were also found upregulated. These data suggest that the amount of miRNA in plasma is also strongly influenced by acute exercise and chronic resistance training [96]. PE is also able to modulate the protein load of muscle EVs. Specifically, using nano-ultra-high-performance liquid chromatography, Whitham and colleagues identify a change in the content of 322 proteins before and after exercise. The presence of glycolytic enzymes within EVs was also found in blood samples of subjects undergoing PE. These results suggest that when work/energy demands are in the growth phase, muscle cells are able to secrete glycolytic enzymes into EVs [21]. Moreover, EVs after PE are enriched with some proteins involved in cell death processes (e.g., programmed cell death 6-interacting protein, Annexin A11, metalloproteinase domain-containing protein 12), inflammation and lipid metabolism, and glucose. The origin of these EVs released after PE can be hypothesized to be muscular, which are discharged into the bloodstream and exert paracrine and endocrine functions [21,96]. The EVs released by the SM can therefore also act over long distances. For example, a single resistance exercise induces an increase in the production of muscle EVs containing miRNA-1, which acts by increasing lipolysis in adipose tissue, determining an increase in lipolysis [31]. Research indicates that, following acute/chronic PE, the concentration of antioxidant enzymes in EVs increases, which in turn plays an inhibitory role in cellular senescence. In particular, it has been demonstrated that superoxide dismutase (SOD1), catalase (CAT), and glutathione peroxidase (GPX) significantly increase EVs after exercise and mediate protection against intracellular oxidative damage [97]. Another important aspect that seems to be regulated by SM-EVS during PE is the capillarization of SM tissue. It affects exercise and performance, maintenance of muscle mass, and insulin response. This process involves communication between endothelial cells and muscle cells: the miRNAs transported by SM-EVs (miRNA-133a, miRNA-206, miRNA-126, miRNA-130a) are transferred to endothelial cells, where they induce pro-angiogenic activation of the nuclear factor κB (NF-κB) [98]. The content of muscle EVs seems to have an important function in counteracting inflammation induced by PE. In obese subjects, 1 week of aerobic/resistance training was found to significantly reduce inflammation by increasing miRNAs in SM-EVs, which target pro-inflammatory and metabolic factors such as interleukin-10, IL-6, macrophage activity, Toll-like receptors, HMGB1, NF-κB, BGβγ, and peroxisome proliferator-activated receptor (PPAR) [99]. The decrease in the inflammatory process is also accompanied by an increase in peroxisome proliferator-activated receptor gamma coactivator 1-alpha (PGC-1α) expression, which induces an increase in lipid metabolism, exercise tolerance, and mitochondrial biogenesis [100]. This process also brings benefits at a systemic level as the EVs release pro-inflammatory cytokines following PE, producing beneficial effects at the brain level by reducing inflammation and preventing neurodegenerative diseases [101].

Numerous studies have explored the effect of exercise on EVs and other biomarkers. Jenkins et al. observed that endurance exercise before a meal reduced the production of reactive oxygen species in circulating CD31+ cells, suggesting a protective effect against oxidative stress [102]. Strohacker et al. found that moderate pre-meal cycling attenuated the postprandial increase in endothelial microparticles (EMP) and monocytic adhesion molecules CD11a and CD18 in young adults, indicating that exercise may mitigate the postprandial inflammatory response [103]. Highton et al. observed that aerobic exercise reduces the percentage of tissue factor-expressing microparticles after consumption of a standardized meal, suggesting that PE may reduce meal-associated thrombotic risk [104]. However, Kirk et al. found a significant increase in endothelial CD105+ and CD106+ microparticles after exercise, with no effect of sodium bicarbonate supplementation, suggesting that intense exercise may increase the release of endothelial EVs as a stress response [105]. Adams et al. found that exercise interventions and weight loss alter miRNA expression in women with breast cancer, highlighting that exercise may modulate gene expression through EVs [106]. Finally, Harrison et al. examined the effect of a high-fat meal on EVs in young, active men, finding a postprandial increase in EVs not attenuated by prior exercise, suggesting that a single episode of exercise may not be sufficient to counteract the acute effects of a high-fat meal [107]. Table 2 summarizes the main features of the reviewed studies investigating the effect of exercise on extracellular vesicles (EV) and microRNA (miRNA) profiles.

To date, further studies are needed to better understand the loading and release mechanisms of muscle EVs triggered by PE. In fact, EVs are excellent vectors that could be used as new therapeutic strategies to improve the effects of PE and counteract the onset of chronic pathologies, such as sarcopenia.

### 6.6. Modulating EVs for Health Benefits

Previous studies have highlighted the significant impact of diet, specific foods, and exercise on the release, composition, and function of EVs. These works show that dietary patterns, such as the MD, and specific foods can modulate EVs, potentially influencing cardiovascular health, metabolic regulation, and aging processes. Furthermore, the role of exercise in altering EVs profiles underlines the complex interplay between lifestyle factors and cellular communication mechanisms. Understanding these relationships is crucial for developing new strategies to harness EVs for the benefit of health.

The MD has been shown to significantly influence EVs profiles. Studies by Chiva-Blanch et al. have shown that this diet can reduce the release of pro-thrombotic microvesicles, thus offering a protective effect against cardiovascular events [68,69]. This suggests that dietary patterns may modulate EVs-mediated cellular communication, which could be exploited as predictive biomarkers of cardiovascular risk. Similarly, Kwon et al. found that the MD alters miRNA expression in EVs of breast cancer survivors, indicating potential benefits beyond cardiovascular health and extending to cancer survival [70].

Specific foods also play a key role in modulating EVs. For instance, oats have been found to reduce inflammatory states in patients with type 2 diabetes, as evidenced by changes in circulating microparticles [73]. Flavanols from cocoa, grapes, walnuts, and blueberries have shown several beneficial effects, such as improving endothelial function and reducing procoagulant microparticles, thus highlighting the potential of these foods in maintaining vascular health and metabolic regulation by modulating EVs [71,74,76,77,78]. These results suggest that supplementation of specific nutrient-rich foods in the diet could improve health outcomes through EVs-mediated pathways.

Dietary supplements have also been studied for their effects on EVs, revealing a significant impact on health. Fish oil supplementation, for example, increases endothelial progenitor cells and reduces microparticles in individuals at cardiovascular risk, indicating improved vascular health [89]. Probiotics such as *Lactobacillus plantarum* have shown promise in modulating EVs to improve skin health and microbial balance [87]. These studies highlight the potential of targeted dietary supplements to influence EV profiles, thus offering new avenues for therapeutic interventions to improve overall health.

Exercise induces significant changes in EVs, which play a crucial role in mediating the beneficial effects of PE. Exercise has been shown to alter miRNA expression and protein content of EVs, and endurance training and resistance training both influence EV profiles [21,102,103,106]. This modulation of EVs by exercise contributes to improved oxidative stress responses, reduced inflammation, and improved metabolic regulation. These findings suggest that regular PE can harness the potential of EVs to promote health and prevent age-related diseases, emphasizing the importance of an active lifestyle to maintain cellular communication and overall well-being.

Despite the growing interest in the role of exercise and diet in modulating EVs, significant limitations persist in the current body of evidence. Many studies remain observational or pre-clinical, with a limited number of large-scale randomized controlled trials exploring the specific effects of exercise and dietary interventions on EV release and function. Furthermore, the heterogeneity of methodologies used for isolating, characterizing, and measuring EVs makes it difficult to compare results between studies. Moreover, the variability in the types of exercise, intensities, and nutritional interventions tested contributes to the challenge of establishing consistent and reproducible results. Finally, although preliminary studies suggest potential therapeutic applications of EVs, further research is needed to clarify their role as biomarkers or therapeutic targets, particularly in human populations and under different physiological conditions.

In summary, Figure 4 illustrates how various nutritional interventions, including specific foods, nutrient-rich diets, and supplements, modulate EV content and function, potentially contributing to improved muscle health and overall metabolic function.

## 7. Conclusions

EVs play a crucial role in both the function and dysfunction of SM. They are involved in intercellular communication, influencing various physiological and pathological processes within the muscle and other tissues. The biogenesis, characterization, and function of EVs in SM are complex and depend on numerous factors, including PE and nutritional interventions. Regular PE and adequate nutrition have been shown to modulate the release and content of EVs, thereby affecting muscle health, metabolism, and overall well-being. In our review, we discussed the role of EVs and their content at the SM level. EVs and their components are able to modulate the homeostasis and metabolism of SM both in physiological and pathological conditions, as in the case of sarcopenia. Furthermore, in recent years the role of EVs in mediating therapeutic effects induced by PE has emerged. Despite this, to date, it is not yet clear whether the EVs released by the SM can be used as biomarkers following physical exercise. Thus, further research is crucial to fully understand the mechanisms by which EVs contribute to muscle function and to explore their potential as therapeutic targets for conditions such as sarcopenia and other muscle- related diseases. Integrating both dietary and exercise interventions appears to be a promising approach for optimizing the release and functionality of skeletal muscle-derived EVs. Future research should focus on conducting RCTs that explore the effects of different types of exercise interventions (aerobic, resistance, and combined) on the release, content, and function of skeletal muscle-derived EVs. These studies should use standardized methodologies for the isolation and characterization of EVs to improve comparability between studies. Furthermore, nutritional interventions should examine specific diets, such as the Mediterranean diet or those rich in polyphenols, omega-3 fatty acids, and specific vitamins, to determine their influence on EV content and function. Studies should measure outcomes such as strength and muscle mass, endurance, and metabolic biomarkers, together with a detailed profile of EVs (miRNAs, proteins, and lipids). It is also essential that future studies employ advanced EV profiling techniques and adopt long-term follow-up designs to better assess the sustained effects of exercise and diet on muscle health. Future investigations should also focus on defining EVs as reliable biomarkers or therapeutic targets in clinical settings, particularly in age-related muscle mass loss and metabolic disorders.

## Figures and Tables

**Figure 1 nutrients-16-03097-f001:**
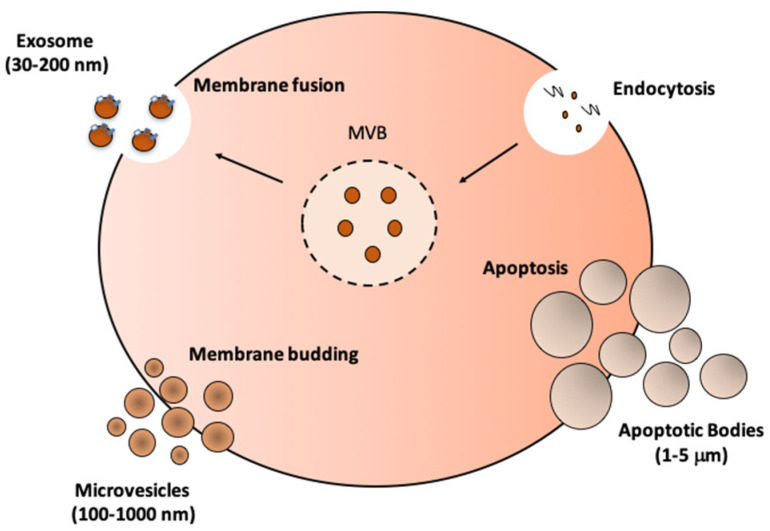
Formation and release of various types of extracellular vesicles (EVs) from a cell.

**Figure 2 nutrients-16-03097-f002:**
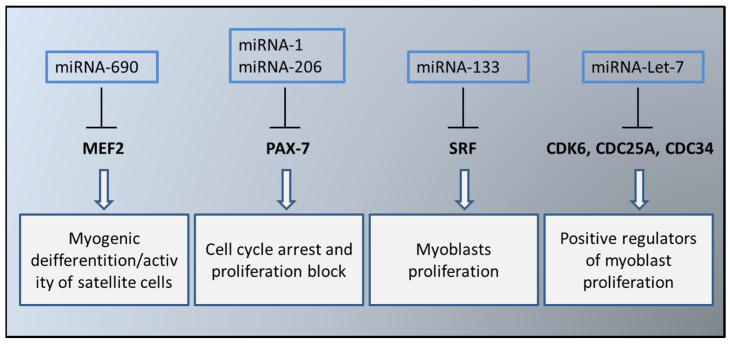
Different roles of some miRNAs implicated in the process of muscle sarcopenia.

**Figure 3 nutrients-16-03097-f003:**
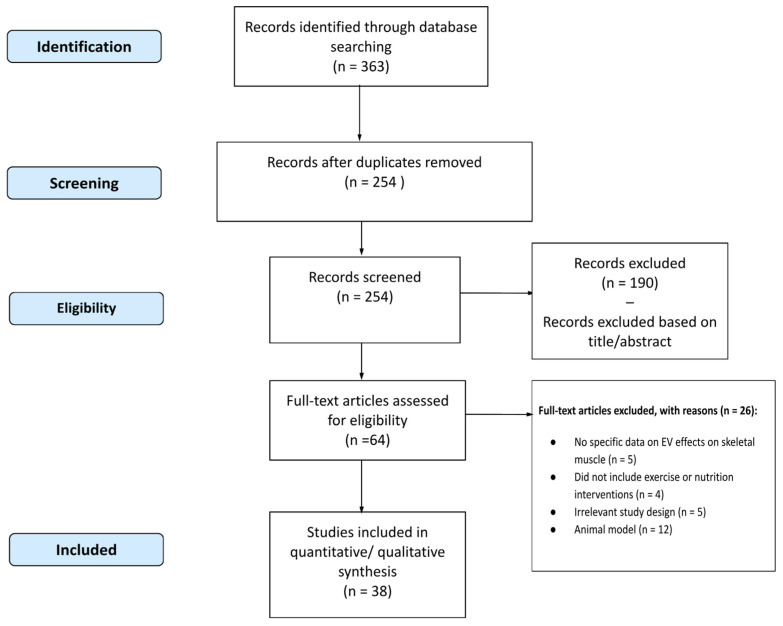
Flowchart of Study Selection Process.

**Figure 4 nutrients-16-03097-f004:**
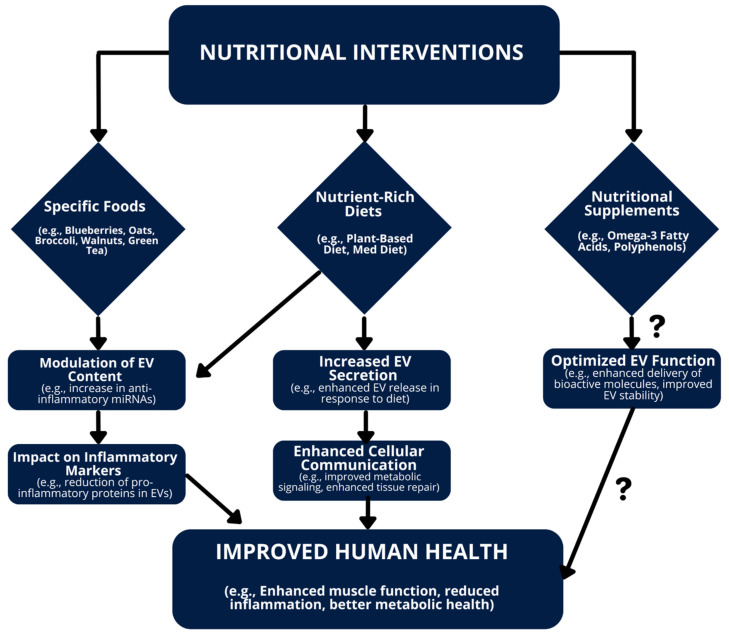
Summary of the impact of nutritional interventions (diets and supplements) on extracellular vesicles (EVs) and their potential role in improving human health. Specific foods, nutrient-rich diets, and supplements influence the content and function of EVs, leading to improved muscle function, reduced inflammation, and improved metabolic health. The question marks (?) indicate areas where further research is needed to fully understand the mechanisms by which nutritional supplements impact EVs function, including how they enhance the delivery of bioactive molecules and improve EVs stability.

**Table 2 nutrients-16-03097-t002:** Summary table of studies on the effect of exercise on extracellular vesicles and miRNA profiling.

First Author	Year	Sample Size (n)	Study Design	Population	Duration	Intervention Type	Effects on EV	Other Outcomes	Ref.
Harrison, M.	2009	8	Clinical Trial	Recreationally active young men	Single session	High-fat meals, exercise	EMP ↑ postprandially, no attenuation by prior exercise	↓ postprandial triglycerides, ↑ HDL-C, no changes in sICAM-1, sVCAM-1, IL-6, or leukocytes	[107]
Jenkins, N. T.	2011	10	Controlled Clinical Trial	Healthy men	Single session	Endurance exercise	Prevented postprandial lipaemia-induced ↑ in ROS in CD31+ cells	↑ antioxidant gene expression, ↓ intracellular lipid uptake, lower serum triglyceride and oxidized LDL-cholesterol, lower plasma endothelial microparticle concentrations	[102]
Strohacker, K.	2012	12	RCT	Young adults	Single session	Moderate-intensity premeal cycling	↑ in EMP and CD11a/CD18 monocyte cell surface receptors	Exercise ↓ postprandial monocyte and endothelial cell activation	[103]
Kirk, R. J.	2013	7	RCT	Healthy male volunteers	Single session	Sprint cycling, sodium bicarbonate	↑ in endothelial CD105+ and CD106+ MPs post-exercise, no effect of supplementation	Endothelium rapidly recovers post-exercise in healthy individuals	[105]
Nielsen, S.	2014	32 (13 acute, 7 chronic)	Clinical Trial	Healthy, trained men	12 weeks + single session	Acute endurance exercise and chronic endurance training	Acute: ci-miRNAs downregulated immediately post-exercise, followed by upregulation at 1 and 3 h. Chronic: 7 ci-miRNAs ↓ and 2 ci-miRNAs ↑ post-training	Identified dynamic plasma miRNA changes in response to both acute exercise and chronic endurance training	[108]
Uhlemann, M.	2014	58 (13 + 12 + 22 + 11)	Comparative Study	Healthy adults	Four different exercise protocols	Maximal symptom-limited exercise test, 4-h cycling, marathon, resistance training	↑ miRNA-126 after endurance exercises; ↑ miRNA-133 after resistance exercise	Different exercise modalities impact endothelial and muscle cells differently; miRNA-126 linked to endothelial damage	[94]
Fruhbeis, C.	2015	12	Clinical Trial	Healthy, physically active men	Single session	Incremental cycling and treadmill running until exhaustion	Significant ↑ in small EVs (100–130 nm) immediately after exercise, declining within 90 min; treadmill-induced EVs sustained longer	EV release initiated early during exercise, before reaching anaerobic threshold; potential role in exercise adaptation	[96]
Guescini, M.	2015	22	Clinical Trial	Sedentary and fit young men	Cross-sectional	Physical exercise	SGCA+ EVs enriched for miR-206; correlation between fitness and muscle-specific miRNAs	EV miR-133b and miR-181a-5p significantly upregulated after acute exercise; role in muscle communication	[92]
Adams, B. D.	2018	121	Clinical Trial	Breast cancer survivors	6 months	Exercise, weight loss	Identified eight miRNAs associated with BMI and weight loss interventions, including miR-191-5p and miR-122-5p	Correlated miRNAs with biological pathways such as “Estrogen-mediated S-phase entry” and “Molecular mechanisms of cancer”	[106]
Whitham, M.	2018	10	Clinical Trial	Healthy humans	Single session	1-h cycling exercise	Increase in over 300 EV-contained proteins; localization in the liver	Identified new candidate myokines released into circulation independently of classical secretion	[21]
Highton, P. J.	2019	15	RCT	Healthy men	Single session (x3)	Aerobic exercise, meal consumption	↓ tissue factor (TF) expression on platelet and neutrophil-derived microparticles (MPs) after exercise		[104]
Nair, V.D.	2020	10	Clinical Trial	Sedentary and trained older men	Single session (3 points: Pre, Post, 3hPost)	Aerobic exercise (cycle ergometer)	Baseline: ↑ miR-486-5p, ↑ miR-215-5p, ↑ miR-941, ↓ miR-151b. Acute exercise: Distinct exomiRs in trained vs sedentary groups	IGF-1 signaling pathway regulation differs by training status; potential role in counteracting anabolic resistance	[93]
Sullivan, B.P.	2022	16	Clinical Trial	Sedentary lean and obese adults (8 lean, 8 obese)	7 days	Concurrent aerobic and resistance exercise training	Obesity alters small EV miRNAs targeting inflammatory and growth pathways; exercise training induces anti-inflammatory changes in EVs	↓ IL-8 and Jun mRNA after training; exercise-induced EV miRNAs target pathways related to inflammation and growth	[100]
Kargl, C.K.	2024	18	Clinical Trial	Healthy, recreationally active men and women (18–36 years)	12 weeks	Concurrent resistance and endurance exercise training (CET)	AHRET ↑ EV abundance in trained men only; sex-specific differences in miRNA contents	Predicted regulation of hypertrophy and growth pathways in men more than women	[95]

This table summarizes the studies examining the effect of exercise on extracellular vesicles (EV) and microRNA (miRNA) profiles. ‘The arrows ‘↑’ and ‘↓’ indicate an increase or decrease in levels respectively, while the symbol ‘=’ indicates no significant change.’ Abbreviations used in the table: AHRET: acute heavy resistance exercise test; BMI: body mass index; CET: concurrent exercise training; ci-miRNA: circulating microRNA; EMP: endothelial microparticles; EV: extracellular vesicles; HDL-C: high-density lipoprotein cholesterol; IGF-1: insulin-like growth factor 1; IL-6: interleukin-6; mRNA: messenger RNA; RCT: randomized controlled trial; ROS: reactive oxygen species; SGCA: alpha-sarcoglycan; sICAM-1: soluble intercellular adhesion molecule-1; sVCAM-1: soluble vascular cell adhesion molecule-1; TF: tissue factor; ExomiRs: exosomal miRNAs.

## Data Availability

No data were generated or analyzed in this study, as it is a review of existing literature.

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
