# Peer review of "Functional Role of Extracellular Vesicles in Skeletal Muscle Physiology and Sarcopenia: The Importance of Physical Exercise and Nutrition"

_nutrients, 2024, doi:10.3390/nu16183097_

Round 1

Reviewer 1 Report

Comments and Suggestions for Authors

In this study, the authors aimed to summarize the evidences related to the functional role of skeletal muscle-derived extracellular vesicles in sarcopenia, and the potential impact of physical exercise and nutrition. Overall, the subjects is of interest. However, the manuscript is too long which makes it difficult to read and in some parts lack of integrity. I would suggest to shorten the manuscript with the essential information related to the topic.

1) In some parts EV is referred to as electric vehicles, which does not seem right.

2) In page 2, the authors mention plant-derived electric vehicles, but it does not appear in anywhere else in the manuscript.

3) For methodology, the authors should add a schematic diagram to show the process more clearly.

4) Part 3 through part 5 is out of scope and should be mentioned only briefly if needed. The definition of extracellular vesicle and how various types can be classified in skeletal muscle is only what is needed as a background information.

5) How is part 6 different from part 8.4? The scope of the manuscript as I understand is how exercise regulates EVs in skeletal muscle, and these parts seem as duplicate.

6) The authors should note that in some cases EVs are detected and analyzed in the blood level not skeletal muscle, and therefore has limitation to assume as 'skeletal muscle-derived'. Therefore, the authors should specify where applicable whether the evidence is from circulating level or muscle.

7) Part 7 shows how in pathological condition muscle EVs are regulated, and part 8 shows how nutritional and exercise interventions can benefit. These two sections are only the ones needed to match with subject.

Author Response

Dear Reviewer,

First of all, we would like to thank you for the valuable impulses that allowed us to improve the quality of the manuscript. All changes made are highlighted by yellow color, in the revised version of the manuscript, to facilitate the review process. Hoping that we have satisfied your requests as much as possible, we kindly ask you to re-evaluate our paper. 

The Authors

Reviewer 1

In this study, the authors aimed to summarize the evidence related to the functional role of skeletal muscle-derived extracellular vesicles in sarcopenia, and the potential impact of physical exercise and nutrition. Overall, the subject is of interest. However, the manuscript is too long which makes it difficult to read and in some parts lack of integrity. I would suggest to shorten the manuscript with the essential information related to the topic.

1) In some parts EV is referred to as electric vehicles, which does not seem right. 

Thank you for pointing out the error. We have revised the manuscript and corrected the two occurrences where ‘EV’ was misinterpreted as ‘electric vehicles’. Now, throughout the text, ‘EV’ is correctly used to refer to ‘extracellular vesicles’.

2) In page 2, the authors mention plant-derived electric vehicles, but it does not appear anywhere else in the manuscript. 

  1. We have now specified the discussion relating to plant-derived EVs (PDEVs).

3) For methodology, the authors should add a schematic diagram to show the process more clearly.

Thank you for your comment. We have integrated a flowchart illustrating the selection process of the studies included in the review, as per your request. We hope that this addition will make the methodological pathway used clearer.

4) Part 3 through part 5 is out of scope and should be mentioned only briefly if needed. The definition of extracellular vesicles and how various types can be classified in skeletal muscle is only what is needed as a background information. 

OK thanks. We have now summarized paragraphs 3 to 5 keeping only the necessary information.

5) How is part 6 different from part 8.4? The scope of the manuscript as I understand is how exercise regulates EVs in skeletal muscle, and these parts seem as duplicate.

  1. We have now eliminated paragraph 8.4.

6) The authors should note that in some cases EVs are detected and analyzed in the blood level not skeletal muscle, and therefore have limitations to assume as 'skeletal muscle-derived'. Therefore, the authors should specify where applicable whether the evidence is from circulating level or muscle.

Thank you for your comment. We have revised the manuscript and specified, where applicable, whether the evidence reported relates to EVs detected in circulation or directly in skeletal muscle. We are aware of the limitations in assuming that all EVs detected in blood are derived from skeletal muscle, and have clarified this in the relevant places in the text. We greatly appreciate your comment and hope that the changes we have made will meet your expectations.

7) Part 7 shows how in pathological condition muscle EVs are regulated, and part 8 shows how nutritional and exercise interventions can benefit. These two sections are only the ones needed to match with the subject.

We thank you for your comments. We have taken your feedback on board and revised the manuscript focusing only on those aspects most relevant to the topic at hand. In particular, we have retained and optimised sections 7 and 8, which deal respectively with the regulation of muscle EVs under pathological conditions and the effectiveness of nutritional interventions and exercise. These sections are those most relevant to the subject of the article, as per your indication.

Reviewer 2 Report

Comments and Suggestions for Authors

This review is very interesting and appropriate for publication. However, the following additional information should be provided for readers.

1. It is desirable to give the brief overview regarding the relationship of EVs with age-related chronic-diseases such as cardiovascular diseases, diabetes and cancers. 

2. The brief overview of the relationship of EVs with inflammation & oxidative stress also should be added.

3. What is the figure “nutrients-3152062-non-published pdf (Dietary Influences on Adiponectin)”?

Author Response

Dear Reviewer,

First of all, we would like to thank you for the valuable impulses that allowed us to improve the quality of the manuscript. All changes made are highlighted by yellow color, in the revised version of the manuscript, to facilitate the review process. Hoping that we have satisfied your requests as much as possible, we kindly ask you to re-evaluate our paper. 

The Authors

Reviewer 2

This review is very interesting and appropriate for publication. 

We sincerely thank the reviewer for the positive feedback and for considering our work interesting and suitable for publication. We are pleased that our contribution was appreciated and remain at your disposal for any further suggestions.

However, the following additional information should be provided for readers.

  1. It is desirable to give the brief overview regarding the relationship of EVs with age-related chronic-diseases such as cardiovascular diseases, diabetes and cancers. 

Thank you for your valuable feedback. We have now added the part about the relationship between EVs and chronic diseases (cardiovascular disease, diabetes and cancer) in the first paragraph of the introduction.

  1. The brief overview of the relationship of EVs with inflammation & oxidative stress also should be added.

Done. We have now added the role of EVs in inflammation and oxidative stress in the first paragraph of the introduction.

  1. What is the figure “nutrients-3152062-non-published pdf (Dietary Influences on Adiponectin)”?

We apologise for the error. The image you refer to, ‘nutrients-3152062-non-published pdf (Dietary Influences on Adiponectin)’, was uploaded by mistake and is related to another paper unrelated to this manuscript. We have removed the link and updated the content correctly. Thank you for reporting the problem and we apologise again for the inconvenience.

Reviewer 3 Report

Comments and Suggestions for Authors

The review discusses the effect of exercise and diet on estracellular vesicles from muscle. Overall, I think the review is a bit disorganized and doesn’t place enough focus on nutrition. I think it might be better to organize the manuscript based on specific disease conditions (the section on sarcopenia for example is okay) and then mention how different dietary components might affect EVs and the disease conditions.

Specific comments:

Abstract, line 26: “electric vehicles” – this should be changed to “extracellular vesicles”

Line 38: I suggest inserting a period and starting a new sentence at “In fact”

Line 40: “For these reasons, EVs are now used in multiple types of antiinflammatory therapy, in eating disorders...” It would be beneficial to have a reference at the end of this sentence.

Line 56: “Moreover, in recent years, interest has emerged in plant-derived electric vehicles (PDEVs), which, containing molecules of antioxidant origin, are considered therapeutic agents against many pathologies including muscular ones” Again, it would be of benefit to the reader if you could include references at the end of this sentence.

Last paragraph of the introduction: Here you mention you will be discussing the role of nutrition, but throughout the introduction, I don’t think you’ve mentioned nutrition at all. It would be ideal to give your reader a quick background / summary of the role of nutrition in the introduction.

Line 80: It is stated that muscle is considered the body’s largest organ. I believe the skin is most often considered the body’s largest organ.

Line 86: add the word “when” before “released”

Paragraph starting line 80: I think this paragraph might stray from the topic of extracellular vesicles, with its discussion of cytokines and myokines, unless these are delivered in extracellular vesicles.

Line 133: “Alongside miRNAs in muscle EVs, tRNAs have also been identified, the function of which is not yet known.” Again, a reference here would be helpful.

Line 156: References are needed to support the statement in the first couple of sentences of this paragraph.

I am not sure Table 1 is very useful in the paper. To me, the paper should focus on the physiology of EVs and not the methodology of how to extract them. I suggest deleting the table.

Same comment for the paragraph starting on line 188 and Tabe 2.

Line 296: I suggest changing “genders” to “sex”

Line 474: This is the first mention of diet in the review. I think for a comprehensive review article in
Nutrients you need to integrate the effects of diet throughout each section.

In sections 8.3 and 8.4 you summarize the effects of diet and exercise on EVs. These paragraphs tend to mention one study after another in separate sentences. I think these paragraph need to be better organized, rather than simply stating the effects of random studies one after another.

Comments on the Quality of English Language

The English needs minor revision

Author Response

Dear Reviewer,

First of all, we would like to thank you for the valuable impulses that allowed us to improve the quality of the manuscript. All changes made are highlighted by yellow color, in the revised version of the manuscript, to facilitate the review process. Hoping that we have satisfied your requests as much as possible, we kindly ask you to re-evaluate our paper. 

The Authors

Reviewer 3

The review discusses the effect of exercise and diet on extracellular vesicles from muscle. Overall, I think the review is a bit disorganized and doesn’t place enough focus on nutrition. I think it might be better to organize the manuscript based on specific disease conditions (the section on sarcopenia for example is okay) and then mention how different dietary components might affect EVs and the disease conditions.

Specific comments:

Abstract, line 26: “electric vehicles” – this should be changed to “extracellular vesicles”

Thank you for pointing out the error. We have revised the manuscript and corrected the two occurrences where ‘EV’ was misinterpreted as ‘electric vehicles’. Now, throughout the text, ‘EV’ is correctly used to refer to ‘extracellular vesicles’.

Line 38: I suggest inserting a period and starting a new sentence at “In fact”

Thank you for your suggestion. We have made the change by inserting a period and starting a new sentence with 'In fact,' as you indicated. 

Line 40: “For these reasons, EVs are now used in multiple types of antiinflammatory therapy, in eating disorders...” It would be beneficial to have a reference at the end of this sentence.

  1. We have now added the reference.

Line 56: “Moreover, in recent years, interest has emerged in plant-derived electric vehicles (PDEVs), which, containing molecules of antioxidant origin, are considered therapeutic agents against many pathologies including muscular ones” Again, it would be of benefit to the reader if you could include references at the end of this sentence.

  1. We have now added the reference.

Last paragraph of the introduction: Here you mention you will be discussing the role of nutrition, but throughout the introduction, I don’t think you’ve mentioned nutrition at all. It would be ideal to give your reader a quick background / summary of the role of nutrition in the introduction.

Thank you for your valuable feedback. We have revised the introduction to include a brief background on the role of nutrition in relation to extracellular vesicles (EVs). Specifically, we have highlighted how dietary interventions can influence the production, composition, and function of EVs, which is relevant for understanding the interaction between diet, exercise, and cellular communication.

Line 80: It is stated that muscle is considered the body’s largest organ. I believe the skin is most often considered the body’s largest organ.

Thank you for your insightful comment. You are correct that the skin is generally considered the body’s largest organ in terms of surface area. To avoid confusion, we have revised the sentence to clarify that skeletal muscle is one of the largest organs in the body by weight, representing approximately 40% of total body mass.

Line 86: add the word “when” before “released”

Done.

Paragraph starting line 80: I think this paragraph might stray from the topic of extracellular vesicles, with its discussion of cytokines and myokines, unless these are delivered in extracellular vesicles.

Thank you for your observation. We have revised the paragraph to ensure it remains focused on the role of extracellular vesicles (EVs) in skeletal muscle communication. We have clarified that many of the myokines, cytokines, and exerkines discussed are indeed delivered through EVs, which aligns with the primary topic of this review.

Line 133: “Alongside miRNAs in muscle EVs, tRNAs have also been identified, the function of which is not yet known.” Again, a reference here would be helpful.

  1. We have now added the reference.

Line 156: References are needed to support the statement in the first couple of sentences of this paragraph.

  1. We have now added the references.

I am not sure Table 1 is very useful in the paper. To me, the paper should focus on the physiology of EVs and not the methodology of how to extract them. I suggest deleting the table. Same comment for the paragraph starting on line 188 and Table 2.

Done. Thank you for your suggestion. We have now deleted the Table 1 and Table 2. However, we believe the content was important to highlight, so we have summarized the key points within the text to ensure the essential information is still conveyed.

Line 296: I suggest changing “genders” to “sex”

Done.

Line 474: This is the first mention of diet in the review. I think for a comprehensive review article in Nutrients you need to integrate the effects of diet throughout each section.

Thank you for the feedback. We have integrated the effects of diet throughout the main sections of the manuscript:

Introduction: We emphasised the role of nutritional interventions in modulating EVs and their interaction with exercise.

Skeletal Muscle as an Endocrine Tissue: We added a brief discussion on the influence of specific dietary components, such as omega-3 fatty acids and polyphenols, in the composition of EVs and endocrine function of muscle.

Biogenesis and Characterization of EVs: We included how diet may influence the biogenesis and lipid composition of EVs, highlighting the role of vitamins and polyunsaturated fats.

Sarcopenia and Muscle EVs: We expanded the discussion on nutritional interventions, highlighting diets rich in antioxidants and anti-inflammatory nutrients.

Conclusion: We emphasised the importance of the synergy between diet and exercise in optimising the function of muscle EVs.

We hope that these modifications adequately address your suggestions.

In sections 8.3 and 8.4 you summarize the effects of diet and exercise on EVs. These paragraphs tend to mention one study after another in separate sentences. I think these paragraph need to be better organized, rather than simply stating the effects of random studies one after another.

Thank you for your suggestion regarding sections 8.3 and 8.4. We have revised and reorganised the sections to ensure a more coherent and flowing structure. Instead of listing the studies sequentially, we have reorganised the information thematically, grouping the studies by category and discussing the main mechanisms related to diet and exercise. In addition, we have separated the original table into two separate tables, one for the effects of diet and one for the effects of exercise, to provide a clearer, more structured view of the data and make it easier to consult. These changes should make the sections more organic and improve overall readability.

Round 2

Reviewer 3 Report

Comments and Suggestions for Authors

The authors have responded adequately to my comments

Author Response

Thank you